# MET-Bench: Multimodal Entity Tracking for Evaluating the Limitations of Vision-Language and Reasoning Models

**Vanya Cohen** [1]   **Raymond Mooney** [1]

## Abstract

Entity state tracking is a necessary component of world modeling that requires maintaining coherent representations of entities over time. Previous work has benchmarked entity tracking performance in purely text-based tasks. We introduce MET-Bench, a multimodal entity tracking benchmark designed to evaluate the ability of vision-language models to track entity states across modalities. Using three domains, we assess how effectively current models integrate textual and image-based state updates. Our findings reveal a significant performance gap between text-based and image-based entity tracking. We empirically show this discrepancy primarily stems from deficits in visual reasoning rather than perception. We further show that explicit text-based reasoning strategies improve performance, yet limitations remain, especially in long-horizon multimodal tasks. We apply reinforcement learning to improve entity tracking in open-source VLMs. This yields substantial in-modality gains, but does not transfer robustly across input modalities. Our results highlight the need for improved multimodal representations and reasoning techniques to bridge the gap between textual and visual entity tracking.

## 1. Introduction

Developing models that track and predict the latent state of the world has emerged as an important goal for artificial intelligence (Ding et al., 2025). One capability of effective world models is entity state tracking: estimating the evolution of entities, their attributes and relations over time. Many problems require accurate entity tracking, including robotic manipulation (Yoneda et al., 2024; NVIDIA

et al., 2025), video question-answering (He et al., 2025), and computer-control agents (Bishop et al., 2024; Sager et al., 2025). Success on these tasks requires maintaining a coherent estimate of how entity state changes from sequential, multimodal observations.

Early progress on entity tracking emerged within text-only tasks. From coreference resolution (Hobbs, 1978; Lappin & Leass, 1994) and discourse processing to narrative comprehension, computational linguistics has long grappled with the challenge of maintaining coherent entity representations across textual contexts (Bunescu & Paşca, 2006; Schank & Abelson, 1977).

While significant progress has been made in text-only tasks (Liu et al., 2023; Papalampidi et al., 2022), the broader challenge of understanding how entities change through sequences of actions or events remains an open challenge (Fagnou et al., 2024; Kim & Schuster, 2023; Toshniwal et al., 2022). This limitation becomes apparent in tasks requiring integration of information across multiple modalities, an increasingly important frontier in AI as systems are asked to reason about content that combines text with other modalities like images and video.

Our work examines this challenge through the lens of multimodal entity state tracking, where changes to entity states must be understood from both textual descriptions and visual observations. This setting provides a natural extension to classical NLP problems while also connecting to emerging research in multimodal dialogue, robotics, and human-AI interaction. We focus specifically on scenarios where multimodal models must reason about world-state changes described through a combination of text and images.

To systematically evaluate models' capabilities in this multimodal setting, we study three domains: *multimodal Chess*, *Shell Game*, and *Minecraft*. Through Chess and Shell Game, we assess how effectively current language models can track entity states when updates are conveyed through both text and images. Our analysis reveals substantial disparities in how models process text-based and image-based entity-state updates, highlighting fundamental limitations in their multimodal understanding. While Chess and the Shell Game provide structured testbeds for evaluating entity tracking,

[1]Department of Computer Science, The University of Texas at Austin, USA. Correspondence to: Vanya Cohen <vanya@utexas.edu>.

*Proceedings of the 43$^{rd}$ International Conference on Machine Learning*, Seoul, South Korea. PMLR 306, 2026. Copyright 2026 by the author(s).

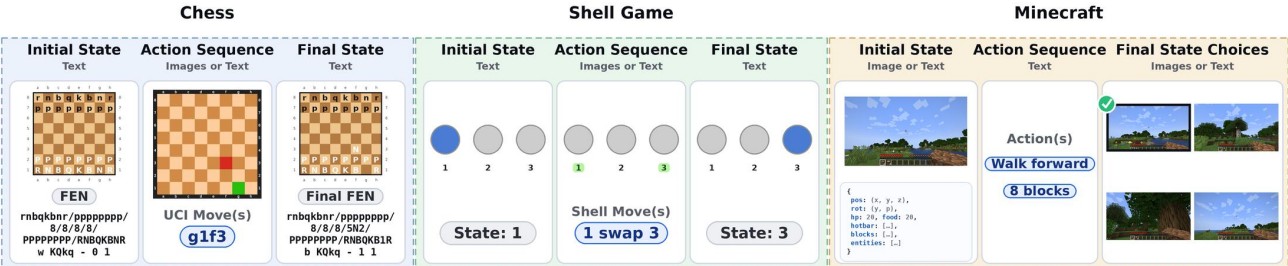

*Figure 1.* In the multimodal entity tracking benchmark (MET-Bench), a vision-language model (VLM) predicts the final entity state from the initial state and actions that update the entity state. In Chess and Shell Game, the initial entity state is provided as text and the actions are given as images or text. The predicted final state is a text representation of the entity state. In Minecraft, the initial state is given as text or image and the actions as text. The task is to predict the correct final state from a multiple choice candidate set of text or image representations.

real-world applications often involve more ambiguous and dynamic environments. However, by isolating state-tracking performance in controlled settings, we establish a clear baseline for assessing multimodal reasoning, one that provides a straightforward means of evaluation where difficulty is easily controlled.

We also evaluate state prediction performance in Minecraft, which offers a setting more applicable to real-world state tracking performance, while still enabling controlled evaluation. In Minecraft, we instead vary the state modality to analyze the difference in models' abilities to reason about states presented as visual and textual inputs.

Overall, this paper makes the following contributions:

- We introduce the multimodal entity tracking benchmark (MET-Bench) that extends text entity tracking evaluation to the multimodal setting for three domains: *multimodal Chess*, *Shell Game*, *Minecraft*.

- We demonstrate that current models, despite stronger performance on pure text tasks, struggle to maintain accurate entity representations when processing state updates from image inputs.

- Through experiments, we show that these limitations primarily stem from higher-level reasoning challenges rather than perceptual issues.

- We apply reinforcement learning to improve entity tracking and find limited transfer learning between modalities, highlighting a continuing challenge for VLM state tracking.

**Conflict of Interest Disclosure** The author VC is employed as a student researcher at Google DeepMind which develops the Gemini and Gemma models evaluated in this paper.

## 2. Background

We formulate the problem of *multimodal entity tracking* as a sequential state estimation task, where an agent must infer the final state of a system given an initial state and a series of observed actions. Formally, at each time step $t$, the environment is in state $\mathbf{S}_t$, and transitions occur according to an action sequence $\mathbf{A} = (\mathbf{a}_1, \mathbf{a}_2, \ldots, \mathbf{a}_T)$. The agent receives $\mathbf{a}_t$ corresponding to each action, which may be textual $\mathbf{a}_t^{\text{text}}$ or visual $\mathbf{a}_t^{\text{image}}$. The objective is to infer the final state:

$$\mathbf{S}_T = \mathrm{f}(\mathbf{S}_0, \mathbf{a}_1, \mathbf{a}_2, \ldots, \mathbf{a}_T),$$

where f is the function modeling entity state updates.

In Chess and Shell Game, MET-Bench represents the initial and final states of each domain as text but also evaluates models' ability to track entity state changes through images. This approach isolates the multimodal entity tracking challenge by ensuring that models begin and end with well-defined textual representations while processing state transitions visually. This assesses the models' capacity to maintain coherent entity representations across modalities while minimizing confounding errors from perceptual failures, which remain a limitation of current vision-language models (Sharma et al., 2024). In Minecraft, actions are represented as text and the input and final states are represented either as text or images.

### 2.1. Chess

Chess is a well-studied domain for testing entity tracking of deep learning models (Toshniwal et al., 2022). The state $S_t$ represents an 8×8 board configuration expressed in Forsyth–Edwards Notation (FEN), actions correspond to legal chess moves from real games, and action observations consist of either symbolic Universal Chess Interface (UCI) notation or visual board-image descriptions of moves. We likewise adopt chess as an entity tracking testbed where the task is to maintain a correct representation of the board state

across a sequence of moves.

Utilizing real games from Millionbase[1] (Toshniwal et al., 2022), we generate sequences of states and actions (moves) using standard chess notation: UCI for actions and FEN for board states. **Text-Encoded Moves** (each action is provided as a short UCI textual description, e.g., "e2e4" for moving a piece from e2 to e4); **Image-Encoded Moves** each action is accompanied by a rendered image serving as a visual representation of the move; see Fig. 1. The visual action depictions were created to enable high perceptual accuracy, as shown by the results in Table 3. In both cases, the final output is the FEN-encoded location of each piece on the board after a sequence of moves. The dataset includes multiple game trajectories of varying length, capturing a variety of piece types and board states.

## 2.2. Shell Game

Shell Game is a classic demonstration of hidden-state tracking. A ball is placed under one of three cups (or shells), which are then swapped pairwise in succession. The goal is to track which cup currently hides the ball. The state $S_t$ tracks the hidden position of a ball under three shells, and actions correspond to swaps between pairs of shells. Other works have explored text-based shell-game-like domains with varying levels of added complexity (Li et al., 2021; Long et al., 2016; Kim & Schuster, 2023).

Shell Game has a simpler entity-state and action space than Chess. However, while many frontier models have been trained on UCI/FEN-encoded chess games, the Shell Game is, to the best of our knowledge, not present in the training data of these models, though there may be analogous tasks in the pretraining data.

We simulate repeated Shell Game swaps to create a set of Shell Game trajectories. Swap actions are either: **Text-Encoded Swaps** "x swap y", where $\{x, y\} \subset \{1, 2, 3\}$; **Image-Encoded Swaps** an image depicting the shells being swapped, with the ball not visible; see Fig. 1. The ground truth entity state after the game finishes is a single number indicating the final shell position of the ball.

## 2.3. Minecraft

Minecraft is a more complex domain where state tracking requires reasoning about partial observation, dynamic environments, and visually complex scenes. The state $S_t$ is the player's local first-person world state, either as an image or textual description including position, orientation, nearby blocks, and visible scene contents. Actions correspond to low-level player inputs such as walking, turning, attacking, using an item, climbing, or descending (e.g., "Walk

forward 8 blocks, attack"). Unlike Chess and Shell Game, where the model predicts a symbolic final state, in Minecraft the model must predict the correct next state from four choices, sampled from adjacent parts of the trajectory. The ground truth output is a single multiple-choice index indicating which candidate is the correct next state.

We collect a dataset of trajectories across multi-step scripted behaviors, including collect wood, explore, and build structure. Each trajectory consists of rendered first-person frames paired with per-tick game telemetry in text. We generate benchmark examples by sampling an input state, the input controls used, and four candidate next states, one of which is the true successor. We evaluate two parallel state encodings: **Text-Encoded States** where the input and candidate states are rendered from game telemetry as structured textual descriptions and **Image-Encoded States** where the input and candidate states are first-person Minecraft screenshots; see Fig. 1. The text-encoded states contain rich task information, which the model must infer from the visual observations.

The text condition serves as an upper bound on task performance: the state information is structured, so accuracy reflects the model's state reasoning ability alone. The image condition presents similar information visually, so any performance gap directly quantifies the deficit introduced by visual state understanding. If a model possessed equal competence in visual and textual reasoning, the two conditions would yield comparable accuracy. The magnitude of this gap therefore isolates the contribution of visual understanding to overall entity tracking failure.

## 2.4. Reinforcement Learning for Reasoning

To improve the entity state reasoning of open-source models, we utilize Group Relative Policy Optimization (GRPO) (Shao et al., 2024), a policy-gradient RL method for training language models. GRPO is suited to tasks with verifiable outcomes, such as entity tracking, where correctness can be determined programmatically.

## 3. Methods

We utilize the standard chat-based input schema exposed by current models that consists of interleaved user-provided and model-generated messages. Figure 2 shows the prompting strategy used for the Chess domain. Similarly, Figure 6 details the prompting strategy used for the Shell Game domain.

In the case of image-action input, the text actions are replaced with their image-rendered versions and a text description of how to interpret the image-actions is provided. Fig. 1 shows the image representations used for the Chess and Shell Game actions. These image representations were

---

[1] rebel13.nl/rebel13

| Role | Messages |
|------|----------|
| User | You are a helpful assistant that tracks chess moves in a game and produces the final FEN. The initial state is: `rnbqkbnr/pppppppp/8/8/8/8/PPPPPPPP/RNBQKBNR w KQkq - 0 1` Here are the moves played: `e2e4` `e7e5` Now what is the final FEN? Output FINAL ANSWER: [FEN]. |
| Assistant | FINAL ANSWER: `rnbqkbnr/pppp1ppp/8/4p3/4P3/8/PPPP1PPP/RNBQKBNR w KQkq - 0 2` |

*Figure 2.* An example zero-shot user–assistant exchange in the **Chess** domain, showing the initial board state as FEN, two UCI moves (e2e4, e7e5) and the final state. For image actions, the UCI moves are replaced with their visual representations and a description of how to interpret these images. The FEN is line-broken for readability.

created through visual prompt engineering to maximize the classification accuracy of actions depicted. Various common image representations were explored including arrows, bounding boxes, and symbolic markers. The image depiction of the game actions is explained to the language model every time images are provided using the prompts shown in the Appendix.

### 3.1. GRPO Training for Entity Tracking

To investigate whether reinforcement learning can improve entity tracking in open-weight vision-language models, we apply GRPO to train models on MET-Bench tasks.

**Reward Function Design** For entity tracking, we use reward functions that leverage the automatic verifiability of our benchmark tasks. For Chess, the reward is the accuracy of the predicted board:

$$R_{\text{chess}}(y, y^*) = \frac{1}{64} \sum_{i=1}^{64} \mathbb{1}[y_i = y_i^*],$$

For Shell Game, the reward is binary based on the ball position:

$$R_{\text{shell}}(y, y^*) = \mathbb{1}[y = y^*].$$

We train Gemma 3 4B IT (Gemma Team et al., 2025). For the Chess and image settings at 10 actions, the base policies do not output valid chain-of-thought; to enable RL training, we initialize the policy with a valid output format by finetuning on synthetic demonstrations. Separately, we find that finetuning on only the final entity state does not generalize, supporting the results in Tables 1 and 2 that sequential state tracking is made easier by generating intermediate reasoning. Full training configurations are in Section A.3.

## 4. Experiments

Our experiments test a wide range of models (Appendix A.5) and settings to evaluate different aspects of frontier-model entity-tracking performance. For all experiments, the models are sampled with a temperature of zero.

### 4.1. Tracking in Text Outperforms Images

We evaluate the difference in accuracy when tracking entity states from text and image actions in zero-shot, few-shot, chain-of-thought, and reasoning settings and report 95% confidence intervals (CI). For the few-shot experiments, $N = 5$ examples are selected at random from the training set and prepended to the test example. To evaluate the effect of chain-of-thought reasoning, we prompt the model as in the zero-shot case but append "think step by step before producing a final answer." In both domains, we evaluate on a set of 500 examples selected at random from the test set, each with a sequence length of ten actions, and 100 for reasoning models.

**Chess** In the Chess domain, we report the per-square accuracy of the predicted board, that is the ratio of correctly predicted pieces (or absence of a piece) to the total number of board tiles. The 'Game Start' baseline is the accuracy of predicting the initial board configuration. After only ten actions, much of the board remains unchanged, so this is a strong baseline.

Table 1 reports the accuracy for text and image actions in the chess domain. Performance in the text modality is significantly better than in the image modality. In the zero-shot setting, the best-performing text model Claude 3.7 Sonnet achieves an impressive $96.1\% \pm 0.2\%$ accuracy, while its image-based counterpart drops sharply to $70.2\% \pm 0.5\%$. A similar trend holds across models, indicating that current models can perform entity tracking in text but fail to integrate visual updates as effectively.

Few-shot prompting, chain-of-thought prompting, and reasoning lead to performance improvements. However, the text tracking accuracy exceeds the image accuracy. Because predicting the initial board position is a strong baseline for the accuracy metric, we report additional metrics (exact FEN match, changed square accuracy, and board legality) to assess the robustness of model predictions in Appendix A.1. The trends for these additional metrics confirm the findings of Table 1. We also perform an ablation using a random mixture of text and image-specified moves in Appendix A.2. As expected, performance drops smoothly as the proportion of image-actions increases.

**Shell Game** The final state is a single number $n \in \{1, 2, 3\}$ that gives the position of the ball, and we measure the accuracy of predicting it. The naive baseline picks

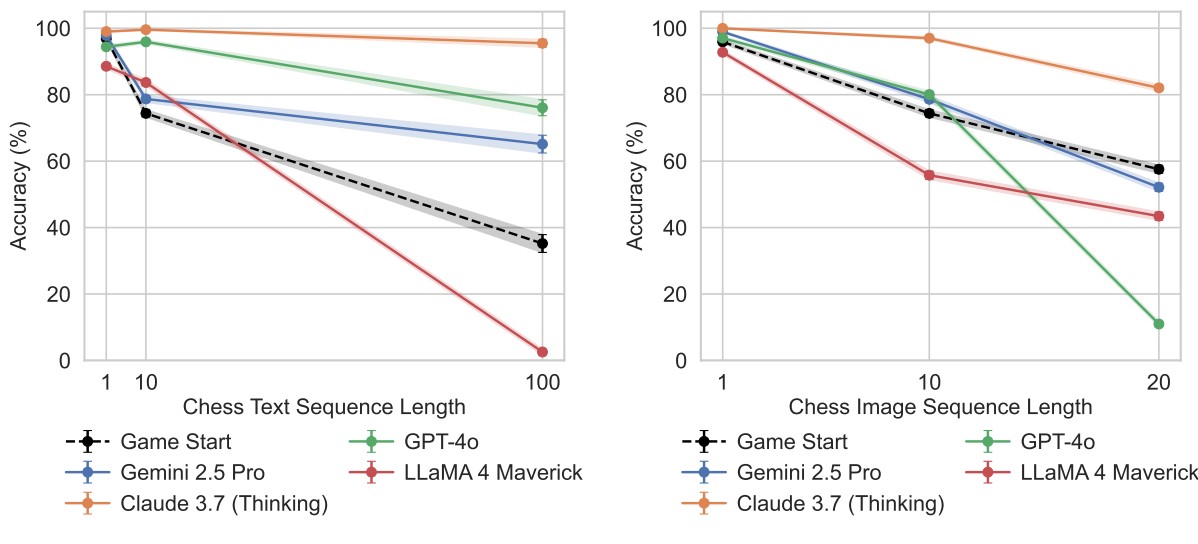

*(a)* Chess accuracy with text actions.

*(b)* Chess accuracy with image actions.

*Figure 3.* In the text action setting, the reasoning models Claude 3.7 Sonnet Thinking and GPT-4o maintain the highest accuracy at longer action-sequence lengths. All models struggle to maintain accurate board representations in the image action setting, with Claude 3.7 Sonnet Thinking performing the best. 95% confidence intervals.

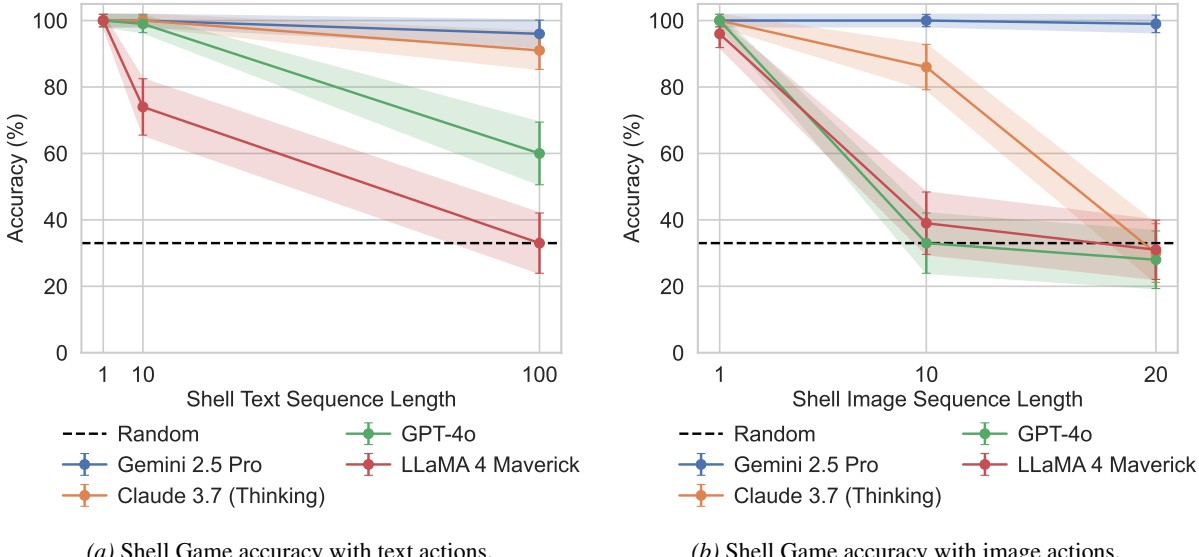

*(a)* Shell Game accuracy with text actions.

*(b)* Shell Game accuracy with image actions.

*Figure 4.* In the text action setting, the reasoning models Claude 3.7 Sonnet Thinking and Gemini 2.5 Pro achieve the highest performance over long action sequences. In the image action setting the accuracy of all models except Gemini 2.5 Pro decreases to random guessing by 20 actions. 95% confidence intervals.

a position uniformly at random.

Table 2 reports the accuracy for text and image actions in Shell Game. The results follow a pattern similar to that of Chess, with text-based tracking outperforming image-based tracking. However, unlike in Chess, the performance in the zero-shot and few-shot settings is close to random.

Chain-of-thought and reasoning lead to large performance increases and multiple models attain near-perfect accuracy

in text. These results suggest that when guided to decompose the task step-by-step, models can reason more effectively (but still largely imperfectly) using image inputs, a finding that complements the results of performing cascaded inference as discussed below.

As in Chess, we perform an ablation using a mixture of text and image-specified moves in Appendix A.2. Unlike in Chess, Shell Game performance drops as the action-modality mixture approaches 50% and then recovers as the

*Table 1.* Entity tracking accuracy (95% CI) in Chess for text-only and image-only actions. In the Few-Shot setting $N = 5$ in-context examples are used. Methods with explicit reasoning perform best. The baseline is predicting the Game Start board configuration.

| MODEL | CHESS | |
| --- | --- | --- |
| | TEXT | IMAGE |
| **BASELINE** | | |
| GAME START | $74.9 \pm 0.5$ | $74.9 \pm 0.5$ |
| **ZERO-SHOT** | | |
| GPT-4O | $91.6 \pm 0.3$ | $76.0 \pm 0.5$ |
| GPT-4O-MINI | $75.6 \pm 0.5$ | $55.3 \pm 0.5$ |
| GPT-4.1 | $85.6 \pm 0.4$ | $67.7 \pm 0.5$ |
| GPT-4.1-MINI | $82.5 \pm 0.4$ | $77.5 \pm 0.5$ |
| GPT-4.1-NANO | $77.5 \pm 0.5$ | $73.3 \pm 0.5$ |
| GEMINI-2.5-FLASH | $91.0 \pm 0.3$ | $66.9 \pm 0.5$ |
| LLAMA-4 MAVERICK | $86.7 \pm 0.4$ | $51.1 \pm 1.2$ |
| MINIMAX-VL-01 | $85.9 \pm 0.4$ | $73.4 \pm 0.5$ |
| CLAUDE 3.7 SONNET | $96.1 \pm 0.2$ | $70.2 \pm 0.5$ |
| **FEW-SHOT (N=5)** | | |
| GPT-4O | $94.3 \pm 0.2$ | $77.6 \pm 0.5$ |
| GPT-4O-MINI | $74.5 \pm 0.5$ | $74.2 \pm 0.5$ |
| GPT-4.1 | $86.6 \pm 0.4$ | $64.9 \pm 0.5$ |
| GPT-4.1-MINI | $81.2 \pm 0.4$ | $76.9 \pm 0.5$ |
| GPT-4.1-NANO | $74.5 \pm 0.5$ | $74.6 \pm 0.5$ |
| GEMINI-2.5-FLASH | $91.3 \pm 0.3$ | $72.0 \pm 0.5$ |
| LLAMA-4 MAVERICK | $85.8 \pm 0.4$ | $48.1 \pm 0.6$ |
| MINIMAX-VL-01 | $88.0 \pm 0.8$ | $40.8 \pm 1.2$ |
| CLAUDE 3.7 SONNET | $99.2 \pm 0.1$ | $77.7 \pm 0.5$ |
| **CHAIN-OF-THOUGHT** | | |
| GPT-4O | $94.2 \pm 0.3$ | $83.4 \pm 0.4$ |
| GPT-4O-MINI | $67.0 \pm 0.5$ | $41.8 \pm 0.5$ |
| GPT-4.1 | $98.1 \pm 0.1$ | $75.3 \pm 0.5$ |
| GPT-4.1-MINI | $86.4 \pm 0.4$ | $77.6 \pm 0.5$ |
| GPT-4.1-NANO | $49.0 \pm 0.6$ | $34.6 \pm 0.5$ |
| GEMINI-2.5-FLASH | $77.0 \pm 1.0$ | $44.6 \pm 1.2$ |
| LLAMA-4 MAVERICK | $82.4 \pm 0.4$ | $61.9 \pm 0.5$ |
| MINIMAX-VL-01 | $62.3 \pm 0.5$ | $32.8 \pm 0.5$ |
| CLAUDE 3.7 SONNET | $99.5 \pm 0.1$ | $96.2 \pm 0.2$ |
| **REASONING** | | |
| O1 | $98.2 \pm 0.3$ | $83.5 \pm 0.9$ |
| O3 | $\mathbf{99.9 \pm 0.1}$ | $45.5 \pm 1.2$ |
| O4-MINI | $84.2 \pm 0.9$ | $78.0 \pm 1.0$ |
| GEMINI-2.5-PRO | $77.4 \pm 1.0$ | $76.8 \pm 1.0$ |
| GEMINI-2.5-FLASH | $40.2 \pm 1.2$ | $17.4 \pm 0.9$ |
| CLAUDE 3.7 SONNET | $\mathbf{99.8 \pm 0.1}$ | $96.0 \pm 0.5$ |
| CLAUDE 4.5 OPUS | $\mathbf{99.9 \pm 0.1}$ | $\mathbf{99.5 \pm 0.2}$ |
| GPT-5.2 | $98.0 \pm 0.4$ | $\mathbf{99.3 \pm 0.2}$ |

*Table 2.* Entity tracking accuracy (95% CI) in Shell Game for text-only and image-only actions. In the Few-Shot setting $N = 5$ in-context examples are used. Methods with explicit reasoning perform best.

| MODEL | SHELL | |
| --- | --- | --- |
| | TEXT | IMAGE |
| **BASELINE** | | |
| RANDOM | $33.3$ | $33.3$ |
| **ZERO-SHOT** | | |
| GPT-4O | $33.0 \pm 4.1$ | $32.2 \pm 4.1$ |
| GPT-4O-MINI | $33.6 \pm 4.1$ | $32.4 \pm 4.1$ |
| GPT-4.1 | $32.2 \pm 4.1$ | $36.4 \pm 4.2$ |
| GPT-4.1-MINI | $31.4 \pm 4.1$ | $30.8 \pm 4.0$ |
| GPT-4.1-NANO | $31.2 \pm 4.0$ | $30.8 \pm 4.0$ |
| GEMINI-2.5-FLASH | $35.0 \pm 4.2$ | $37.0 \pm 4.2$ |
| LLAMA-4 MAVERICK | $30.8 \pm 4.0$ | $33.2 \pm 4.1$ |
| MINIMAX-VL-01 | $30.8 \pm 4.0$ | $31.2 \pm 4.0$ |
| CLAUDE 3.7 SONNET | $35.4 \pm 4.2$ | $37.8 \pm 4.2$ |
| **FEW-SHOT (N=5)** | | |
| GPT-4O | $33.6 \pm 4.1$ | $32.0 \pm 4.1$ |
| GPT-4O-MINI | $34.8 \pm 4.2$ | $32.2 \pm 4.1$ |
| GPT-4.1 | $34.6 \pm 4.2$ | $33.0 \pm 4.1$ |
| GPT-4.1-MINI | $36.4 \pm 4.2$ | $32.8 \pm 4.1$ |
| GPT-4.1-NANO | $36.6 \pm 4.2$ | $36.4 \pm 4.2$ |
| GEMINI-2.5-FLASH | $31.4 \pm 4.1$ | $35.2 \pm 4.2$ |
| LLAMA-4 MAVERICK | $35.4 \pm 4.2$ | $35.2 \pm 4.2$ |
| MINIMAX-VL-01 | $36.4 \pm 4.2$ | $32.0 \pm 4.1$ |
| CLAUDE 3.7 SONNET | $32.4 \pm 4.1$ | $36.0 \pm 4.2$ |
| **CHAIN-OF-THOUGHT** | | |
| GPT-4O | $98.2 \pm 1.2$ | $36.6 \pm 4.2$ |
| GPT-4O-MINI | $66.2 \pm 4.1$ | $33.8 \pm 4.1$ |
| GPT-4.1 | $\mathbf{99.8 \pm 0.5}$ | $37.6 \pm 4.2$ |
| GPT-4.1-MINI | $\mathbf{100.0 \pm 0.4}$ | $72.0 \pm 3.9$ |
| GPT-4.1-NANO | $61.8 \pm 4.2$ | $32.2 \pm 4.1$ |
| GEMINI-2.5-FLASH | $94.0 \pm 4.8$ | $34.0 \pm 9.1$ |
| LLAMA-4 MAVERICK | $77.0 \pm 3.7$ | $34.6 \pm 4.2$ |
| MINIMAX-VL-01 | $77.4 \pm 3.7$ | $34.4 \pm 4.2$ |
| CLAUDE 3.7 SONNET | $\mathbf{100.0 \pm 0.4}$ | $77.4 \pm 3.7$ |
| **REASONING** | | |
| O1 | $\mathbf{100.0 \pm 1.9}$ | $92.6 \pm 2.3$ |
| O3 | $\mathbf{100.0 \pm 1.9}$ | $63.0 \pm 9.3$ |
| O4-MINI | $\mathbf{100.0 \pm 1.9}$ | $33.0 \pm 4.1$ |
| GEMINI-2.5-PRO | $\mathbf{100.0 \pm 1.9}$ | $\mathbf{100.0 \pm 1.9}$ |
| GEMINI-2.5-FLASH | $\mathbf{100.0 \pm 1.9}$ | $42.0 \pm 4.1$ |
| CLAUDE 3.7 SONNET | $\mathbf{100.0 \pm 1.9}$ | $87.0 \pm 6.6$ |
| CLAUDE 4.5 OPUS | $\mathbf{100.0 \pm 1.9}$ | $\mathbf{100.0 \pm 1.9}$ |
| GPT-5.2 | $\mathbf{100.0 \pm 1.9}$ | $\mathbf{99.0 \pm 2.6}$ |

proportion of image actions reaches $100\%$, but to a lower accuracy than text-only tracking. Models may struggle more with reasoning over a mixture of modalities in some cases.

### 4.2. Reasoning Improves Long Sequence Accuracy

We vary the sequence length of the actions to quantify the effect of compounding errors on the models. These sequences range from one to 100 actions in the text action modality and one to 20 in the image action modality. This serves to quantify the relative drop-off in performance among models

in the reasoning and chain-of-thought settings.

**Chess** Figure 3a plots model accuracy against increasing sequence lengths of text actions. The models are evaluated in the chain-of-thought setting for sequence lengths of one to 100 text actions. Figure 3b plots model accuracy against increasing sequence lengths of image actions. The models are evaluated in the chain-of-thought setting for sequence lengths of one to 20 image actions. Reasoning models like Claude 3.7 Sonnet Thinking are able to handle longer sequence lengths with a smaller decrease in accuracy. In con-

*Table 3.* Entity tracking and perception accuracy (95% CI) for Chess and Shell Game. Cascaded inference generally recovers the performance of the text setting, showing the main limitation is visual reasoning not perception. **Perception** accuracy is image-action classification accuracy. In **Cascaded** inference, the model first maps each image action to the text representation of the action, then performs tracking as in **Text**. $\Delta$ is **Cascaded** minus **Image**.

| | CHESS | | | | | SHELL GAME | | | | |
| MODEL | PERCEPTION | TEXT | IMAGE | CASCADED | $\Delta$ | PERCEPTION | TEXT | IMAGE | CASCADED | $\Delta$ |
|---|---|---|---|---|---|---|---|---|---|---|
| GPT-4O | $92.9 \pm 0.7$ | $94.2 \pm 0.3$ | $83.4 \pm 0.4$ | $92.8 \pm 0.3$ | +9.4 | $100.0 \pm 0.1$ | $98.2 \pm 1.2$ | $36.6 \pm 4.2$ | $98.2 \pm 1.2$ | +61.6 |
| GPT-4O-MINI | $44.7 \pm 1.4$ | $67.0 \pm 0.5$ | $41.8 \pm 0.5$ | $64.3 \pm 0.5$ | +22.5 | $100.0 \pm 0.1$ | $66.2 \pm 4.1$ | $33.8 \pm 4.1$ | $68.8 \pm 4.0$ | +35.0 |
| GEMMA 3 27B IT | $71.3 \pm 1.3$ | $53.7 \pm 0.6$ | $48.6 \pm 0.6$ | $52.8 \pm 0.6$ | +4.2 | $100.0 \pm 0.1$ | $56.8 \pm 4.3$ | $30.4 \pm 4.0$ | $58.4 \pm 4.3$ | +28.0 |
| MISTRAL SMALL 3.2 24B | $63.9 \pm 1.3$ | $49.5 \pm 0.6$ | $16.3 \pm 0.4$ | $37.4 \pm 0.5$ | +21.1 | $100.0 \pm 0.1$ | $97.4 \pm 1.4$ | $35.2 \pm 4.2$ | $97.2 \pm 1.5$ | +62.0 |
| QWEN3-VL 235B-A22B | $99.8 \pm 0.2$ | $72.1 \pm 0.5$ | $63.0 \pm 4.2$ | $70.3 \pm 0.5$ | +7.4 | $100.0 \pm 0.1$ | $100.0 \pm 0.4$ | $38.8 \pm 4.3$ | $100.0 \pm 0.4$ | +61.2 |
| GPT-5.4-MINI | $99.8 \pm 0.2$ | $92.6 \pm 0.3$ | $88.7 \pm 0.3$ | $93.6 \pm 0.3$ | +4.9 | $100.0 \pm 0.1$ | $94.8 \pm 2.0$ | $32.4 \pm 4.1$ | $91.8 \pm 2.4$ | +59.4 |

trast, the non-reasoning models experience sharp decreases in accuracy after only a few actions.

**Shell Game** In the text modality in Figure 4a, the reasoning models Gemini 2.5 Pro and Claude 3.7 Sonnet Thinking attain the highest accuracies. Gemini 2.5 Pro performs nearly perfectly at a sequence length of 100 actions, where the non-reasoning models' performance is significantly degraded. In the image modality results in Figure 4b, Gemini 2.5 Pro performs better than the other models, which see a rapid decrease in accuracy with sequence lengths longer than ten actions. By 20 actions, the performance of all models except Gemini 2.5 Pro has degraded to random guessing.

The superior performance of reasoning models suggests that structured inference mechanisms, such as test-time search or chain-of-thought, are crucial for maintaining coherent entity states over long sequences. This aligns with prior findings that models trained on structured reasoning tasks (e.g., mathematics, coding) develop stronger implicit state-tracking capabilities (Kim et al., 2024), whereas standard vision-language models struggle to track entities beyond short sequences.

### 4.3. Tracking Primarily Limited by Visual Reasoning

Using the text actions predicted from the images in the image-action classification task, we devise a cascaded approach to test the effect of performing multimodal inference in two separate steps. Given a sequence of image-based observations $a_t^{\text{image}}$, a vision-language model (VLM) first predicts the corresponding text-based action sequence $\hat{a}^{\text{text}} = g(a^{\text{image}})$, where $g$ maps images to text using the same prompt from image evaluation; Figure 8. The text actions are then used for chain-of-thought inference to estimate the final state as $S_T = f(S_0, \hat{a}^{\text{text}})$. This removes the need for the model to perform entity tracking directly in the image modality, isolating the effect of perceptual failures.

To differentiate perception errors from visual-reasoning errors, we evaluate cascaded inference. Table 3 shows the performance of models on predicting the text action represented by each image. While some models fail to accurately perceive the Chess moves, they perceive the Shell Game

actions with perfect accuracy. The performance in this cascaded setting is more similar to the text-action performance, showing that the model has the task-knowledge needed to perform entity tracking in both domains, but cannot reason effectively directly from the image modality. Together, this indicates that perception of the image-actions is not the only limiting factor for effective entity tracking with image inputs, though there is some error contribution from perception error in Chess.

### 4.4. Minecraft Multimodal World Modeling

*Table 4.* Minecraft state-prediction accuracy (95% CI) using chain-of-thought prompting. To complement the other domains, Minecraft evaluates state prediction from **Text State:** serialized game-state observations and **Image State:** first-person rendered views. Both settings use text actions and the answer is selected from four candidates. Each reported model cell is evaluated on 500 examples; the random baseline is 25%.

| MODEL | MINECRAFT | |
| | TEXT STATE | IMAGE STATE |
|---|---|---|
| **BASELINE** | | |
| RANDOM | 25.0 | 25.0 |
| **CHAIN-OF-THOUGHT** | | |
| GEMMA-3 4B IT | $39.4 \pm 4.3$ | $26.6 \pm 3.9$ |
| GEMMA-3 27B IT | $51.6 \pm 4.4$ | $27.8 \pm 3.9$ |
| QWEN3-VL 8B | $49.4 \pm 4.4$ | $28.4 \pm 3.9$ |
| QWEN3-VL 30B | $72.0 \pm 3.9$ | $31.4 \pm 4.1$ |
| QWEN3-VL 235B | $73.0 \pm 3.9$ | $34.2 \pm 4.1$ |
| GPT-4O | $71.0 \pm 4.0$ | $37.0 \pm 4.2$ |
| CLAUDE SONNET 4.6 | $\mathbf{88.4 \pm 2.8}$ | $\mathbf{41.8 \pm 4.3}$ |

Table 4 evaluates the complementary task of Minecraft game state prediction. The model receives a visual or textual game state and action and must predict from four candidates the next state. This task is complementary to Chess and Shell Game in that the multimodal variation lies in the *state representation* rather than the action representation. By measuring the performance difference between text-based tasks and the related but more challenging image-based tasks, we can understand the extent to which visual state representations limit model reasoning compared to their textual equivalents in a more naturalistic domain.

The results in Table 4 confirm the pattern observed in the

structured domains: models are more accurate on text-based inputs than on image-based inputs. Claude Sonnet 4.6 has the highest performance in both settings, reaching $88.4\% \pm 2.8\%$ with text states and $41.8\% \pm 4.3\%$ with image states. Notably, even the best-performing model in the image setting only exceeds the random baseline by a modest margin, underscoring the difficulty of reasoning about state transitions from first-person visual observations.

Scaling trends are evident in text but not image. Among the Qwen3 VL family, accuracy improves with model size in the text setting, rising from $49.4\%$ at 8B parameters to $73.0\%$ at 235B. However, the corresponding gains in the image setting are far more modest, suggesting that simply scaling model parameters provides diminishing returns for visual state reasoning.

### 4.5. GRPO Improves Entity Tracking

As shown in Tables 1 and 2, methods that use chain-of-thought and explicit reasoning training and inference perform better than single-step zero-shot and few-shot methods. RL training is one way to improve the reasoning abilities of models. We train open-source VLMs to improve entity tracking on the 10-action subset of MET-Bench. The limited cross-modal transfer suggests models process entity tracking in text and image through different pathways. Robust multimodal entity tracking may require architectural innovations or training strategies beyond what RL provides.

Table 5 shows that GRPO training yields substantial gains in the trained modality. For example, when trained on text-only Shell Game inputs, Gemma 3 4B IT improves from $37.6\%$ (near random) to $92.2\%$ accuracy. Despite the improvement in single-modality entity tracking, RL reasoning training does not transfer robustly to the held-out modality. In Shell Game, the accuracy on the held-out modality does not improve. In Chess, there is improvement from the model learning to produce more valid FENs, but only to the level predicting the starting game board. This finding suggests that the entity tracking capabilities learned through RL training are modality-specific.

## 5. Discussion

We demonstrate a significant performance gap between text-based and image-based entity tracking across many state-of-the-art vision-language-reasoning models. This disparity persists across Chess, Shell Game, and Minecraft tasks, suggesting a fundamental limitation in current architectures' ability to reason about entity states through visual observations. This finding is surprising given the results showing models can accurately perceive and classify individual visual actions in Shell Game, and to a lesser extent in Chess. When models first translate visual inputs to text before per-

*Table 5.* Entity tracking accuracy (%) after GRPO training for Gemma 3 4B IT on the 10-action setting. GPT-4o-mini and Gemma 3 27B IT are shown for comparison. 95% confidence intervals.

| MODEL | TEXT | IMAGE |
|---|---|---|
| **SHELL GAME** | | |
| GEMMA 3 4B IT (BASE) | $37.6 \pm 4.2$ | $32.8 \pm 4.1$ |
| + GRPO TEXT | $92.2 \pm 2.4$ | $27.4 \pm 3.9$ |
| + GRPO IMAGE | $39.4 \pm 4.3$ | $90.4 \pm 2.6$ |
| GPT-4O-MINI | $66.2 \pm 4.1$ | $33.8 \pm 4.1$ |
| GEMMA 3 27B IT | $56.8 \pm 4.3$ | $30.4 \pm 4.0$ |
| **CHESS** | | |
| GEMMA 3 4B IT (BASE) | $45.4 \pm 0.6$ | $35.9 \pm 0.5$ |
| + GRPO TEXT | $97.1 \pm 0.2$ | $73.3 \pm 0.5$ |
| + GRPO IMAGE | $67.9 \pm 0.5$ | $92.5 \pm 0.3$ |
| GPT-4O-MINI | $67.0 \pm 0.5$ | $41.8 \pm 0.5$ |
| GEMMA 3 27B IT | $53.7 \pm 0.6$ | $48.6 \pm 0.6$ |

forming entity tracking, they achieve performance comparable to pure text-based tracking. This indicates that the models possess the relevant task knowledge and reasoning capabilities, but struggle to apply them directly in the visual domain.

Likewise, the results in Minecraft show that VLMs achieve substantially higher performance on state prediction when the world state is represented as text rather than as a first-person visual observation. Because the text condition provides complete and unambiguous state information, it serves as an upper bound on task performance, reflecting textual reasoning ability alone. The large gap between text and image accuracy and limited improvements from model scaling indicate that visual state understanding remains far from equivalent to textual reasoning.

The effectiveness of chain-of-thought prompting, particularly in the Shell Game domain where it improved Claude 3.7 Sonnet's accuracy from near random to $100.0\%$ for text and $77.4\%$ for images, highlights the importance of explicit reasoning for entity tracking. Current models can perform complex entity tracking when guided to decompose the task into smaller steps, even in novel domains not explicitly present in their training data. Reasoning models also excel at the long-sequence tasks, but no single model attains perfect accuracy on all tasks, and most models display near-random performance on relatively short image sequences. This indicates that regardless of modality, solving MET-Bench tasks remains an open problem.

The RL experiments demonstrate that reasoning training can substantially improve entity tracking within a single modality, with Gemma 3 4B IT improving from near-random performance to $92.2\%$ accuracy on text-based Shell Game tracking. However, the lack of robust cross-modal transfer suggests that current VLM architectures learn modality-

specific representations rather than unified entity tracking capabilities. Achieving robust multimodal entity tracking may require new methods, such as explicit cross-modal alignment objectives or other inductive biases that encourage models to develop modality-agnostic tracking mechanisms.

## 6. Related Work

Various works have directly and indirectly studied entity state tracking, including in multimodal settings. MET-Bench differs from these by studying parallel multimodal entity state tracking tasks in text and image, allowing for analysis of the unique challenges of tracking entities over analogous tasks in both modalities.

**Textual State Tracking** Entity tracking has been extensively studied in textual domains, with a focus on probing and improving LLMs' abilities to maintain representations of entity states. Toshniwal et al. (2022) evaluate chess as an entity tracking domain, employing finetuned models (Radford et al., 2019) to assess performance. Kim & Schuster (2023) examine the impact of model size and finetuning on entity tracking in textual settings similar to our Shell Game domain. Tandon et al. (2020) construct a benchmark for understanding entity state changes in procedural texts.

Several studies explore the implicit representations of entity states in LLMs. Li et al. (2025a) employ interpretability methods to understand how LLMs implement entity tracking. Li et al. (2021) and Long et al. (2016) use semantic probing to reveal that Transformer-based models (Vaswani et al., 2017) capture entity state representations implicitly during textual reasoning. Building on this, Prakash et al. (2024) demonstrate that finetuning language models for entity tracking tasks enhances pre-existing internal mechanisms rather than learning entirely new representations. Li et al. (2023) find that Transformers trained on Othello games form internal representations of the game state.

Efforts to improve textual entity tracking beyond domain-specific finetuning include Fagnou et al. (2024), which establishes theoretical limitations of the Transformer architecture in tracking entities. They propose a novel attention mechanism for entity tracking in Transformers. Gupta & Durrett (2019) train small Transformer-based models for tracking entity state in instructional texts. Kim et al. (2024) investigate how code pretraining improves language models' abilities to track entities in text, while Yoneda et al. (2024) introduce Statler, a prompting method designed to maintain accurate state representations in text-based robotics planning. Concurrent work uses RL for text-state prediction for single-step transitions (Yu et al., 2026); our work uses RL to train multi-step entity state tracking from images and text.

**Multimodal Procedure Understanding** Multimodal procedure understanding evaluates the ability of models to properly sequence images depicting recipe states (Shirai et al., 2022) or to localize or describe object states (Nguyen et al., 2024; Xue et al., 2024). In contrast, MET-Bench requires tracking state through sequential updates from actions specified in parallel text and image modalities.

**World Modeling Evaluation** World modeling evaluations focus on video generation (He et al., 2025; Li et al., 2025b) and prediction tasks (Gao et al., 2025; Bordes et al., 2025) that lack the combination of parallel modalities, sequential tracking, and scalable difficulty in MET-Bench.

**Tracking and Segmentation** Natural language and vision tracking and segmentation benchmarks evaluate spatial acuity in bounding box placement and pixel-level segmentation of objects and entities (Wang et al., 2021; Hu et al., 2023; Carion et al., 2025). MET-Bench is a complementary task that requires predicting the world state after visual action sequences. The Chess and Shell Game domains involve trivial localization of the relevant entities, yet models fail to reason about sequential visual state updates.

## 7. Conclusion

Our findings suggest the primary bottleneck in multimodal entity tracking is not visual recognition but sequential reasoning over visual updates. Unlike text-based representations, which align with the models' training paradigms, visual updates require implicit state reconstruction, a task most current architectures do not perform reliably. No model achieves perfect accuracy in all long-sequence experiments, highlighting the need for improved training of frontier models for entity state tracking. RL training of entity state reasoning is a promising path forward that can improve the performance of open-source models. The lack of robust cross-modal transfer underscores that visual entity tracking requires fundamentally different representations or training objectives. Future work should explore deficits in entity tracking using mechanistic interpretability methods. Additional research directions include multimodal representation learning methods and explicit memory to mitigate this gap. While the two structured domains, Chess and Shell Game, provide a scalable and verifiable testbed, they do not capture the full complexity of real-world multimodal state tracking. The Minecraft domain extends our evaluation to a more naturalistic setting, but remains within the space of game environments. Future work could leverage complex simulated or real-world environments to evaluate entity state tracking. Addressing these challenges will be crucial for developing AI systems capable of robust multimodal reasoning for real-world tasks.

## Impact Statement

As with any work that improves AI reasoning capabilities, advancements in multimodal entity tracking could eventually contribute to more capable autonomous systems. While our benchmark focuses on game domains (Chess, Shell Game, Minecraft), the underlying capabilities are relevant to robotics and other systems employing automated decision-making. We encourage researchers building on this work to consider appropriate safeguards in consequential real-world settings.

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

# A. Appendix

## A.1. Additional Chess Metrics

We present additional metrics for evaluating chess entity tracking in Appendix Table 9. Beyond the per-square accuracy, we consider three complementary measures that provide a more fine-grained picture of model performance. **Exact-match FEN accuracy** requires the model to predict the exact FEN representation, offering the strictest metric on board prediction. **Changed-square accuracy** isolates performance on squares whose contents differ from the starting position, testing whether models genuinely track game dynamics rather than memorizing the initial board layout. **Legal board position** is the fraction of predictions that are legal board positions. Together, these metrics complement per-square accuracy to provide a robust view of entity tracking performance in Chess.

## A.2. Models Struggle to Integrate Mixed Modalities

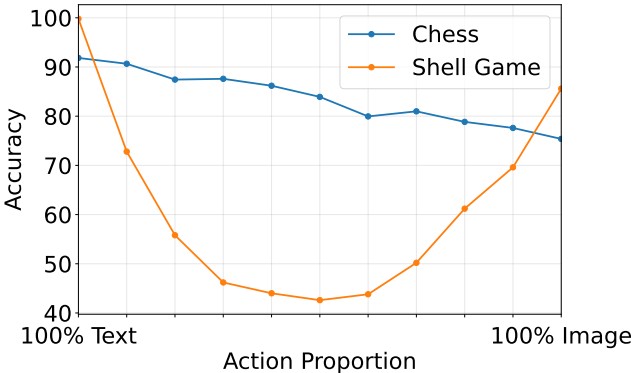

*Figure 5.* Chain-of-thought entity tracking accuracy for Chess and Shell Game with GPT-4o. The data splits range from 100% text-encoded actions to 100% image-encoded actions. The plot illustrates the change in accuracy as actions shift between modalities.

While MET-Bench focuses on parallel, multimodal state tracking tasks, in this ablation we seek to understand how well models can reason about mixed-modality updates. We evaluate performance as we vary the proportion of text and image-based action representations. As shown in Figure 5, for Chess performance degrades smoothly as the fraction of image actions increases, rather than exhibiting an abrupt collapse. However, for Shell Game, the mixture of text and image actions is challenging for the model to reason over.

## A.3. GRPO Training

Appendix Table 6 summarizes the key hyperparameters used for GRPO training across domains and modalities. All experiments are implemented using the TRL library (von Werra et al., 2020) with the Adafactor optimizer (Shazeer & Stern, 2018). Training is performed on a subset of 10,000 10-action trajectories until convergence. For the Chess domain and image modality, the Gemma 3 4B IT base policy is not strong enough to output valid reasoning traces. For this setting, we initialize the policy with finetuning on 1,000 synthetic demonstrations.

*Table 6.* GRPO training hyperparameters.

| | GEMMA 3 4B IT | | GEMMA 3 4B IT | |
| PARAM | CHESS (TEXT) | SHELL (TEXT) | CHESS (IMAGE) | SHELL (IMAGE) |
| --- | --- | --- | --- | --- |
| OPTIMIZER | ADAFACTOR | ADAFACTOR | ADAFACTOR | ADAFACTOR |
| LEARNING RATE | 5E-7 | 1E-7 | 5E-7 | 5E-7 |
| BATCH SIZE | 4 | 4 | 1 | 1 |
| GRAD. ACCUM. | 16 | 16 | 16 | 16 |
| GROUP SIZE | 8 | 8 | 8 | 8 |

| Role | Messages |
|------|----------|
| User | The shell game is a classic game where a ball is hidden under one of three shells. You are a helpful assistant that tracks the position of the ball. The ball starts under shell 2. Here are the moves played:
1 swap 3
2 swap 3
Now what is the final position of the ball? Only output the number 1, 2, or 3. |
| Assistant | 3 |

*Figure 6.* An example zero-shot user–assistant exchange in the **Shell Game** domain, illustrating how the system tracks swaps to determine the ball's final shell.

| Role | Messages |
|------|----------|
| User | You are a helpful assistant that interprets image-based actions in chess.
Here is an image representing a move:
**[Image Input]**
In UCI notation, what move does the arrow on the chessboard represent? The move is from the green square to the red square. (e.g., 'e2e4'). Only output the move and nothing else. |
| Assistant | e2e4 |

*Figure 7.* An example user–assistant exchange in the **Chess** domain, where the assistant identifies the move represented in the image.

## A.4. Benchmark Statistics

We report additional metrics for the benchmark datasets. Table 7 shows the train/validation/test split sizes for each domain.

*Table 7.* Split sizes for the publication datasets. Metrics are reported on a 500 or 100 example subset of the test sets as discussed in Section 4.1.

| SPLIT | EXAMPLES |
|-------|----------|
| **CHESS** | |
| TRAIN | 99,000 |
| VALIDATION | 3,000 |
| TEST | 3,000 |
| **SHELL GAME** | |
| TRAIN | 50,000 |
| VALIDATION | 1,000 |
| TEST | 5,000 |
| **MINECRAFT** | |
| TRAIN | 5,000 |
| VALIDATION | 500 |
| TEST | 1,500 |

## A.5. Model Details

Table 8 details the models used for evaluation and training. Models were selected to sample a wide range of leading providers, including open-source and closed models, and various sizes and protocols for reasoning training.

| Role | Messages |
|------|----------|
| **User** | You are a helpful assistant that interprets image-based actions in the shell game.
Here is an image representing a swap:
**[Image Input]**
In shell game notation, which shells are being swapped in the image? Shells are labeled '1', '2', '3' and the shells being swapped have their numbers highlighted in green. Only output a pair like '1 swap 3' and nothing else. |
| **Assistant** | `1 swap 3` |

*Figure 8.* An example user–assistant exchange in the **Shell Game** domain, where the assistant identifies the shell swap represented in the image.

*Table 8.* Models used in MET-Bench evaluation and training.

| MODEL NAME |
|------------|
| *Anthropic*
CLAUDE 3.7 SONNET (ANTHROPIC, 2025B)
CLAUDE 4.5 OPUS (ANTHROPIC, 2025A)
CLAUDE SONNET 4.6 (ANTHROPIC, 2026) |
| *Google DeepMind*
GEMINI 2.5 FLASH, PRO (GEMINI TEAM, 2025)
GEMMA 3 4B IT, 27B IT (GEMMA TEAM ET AL., 2025) |
| *Meta*
LLAMA 4 MAVERICK (META, 2025) |
| *MiniMax*
MINIMAX-VL-01 (MINIMAX ET AL., 2025) |
| *Mistral*
MISTRAL SMALL 3.2 24B (MISTRAL, 2025) |
| *OpenAI*
GPT-4O, 4O MINI (OPENAI ET AL., 2024A)
GPT-4.1, 4.1-MINI, 4.1-NANO (OPENAI, 2025A)
GPT-5.2, 5.4-MINI (SINGH ET AL., 2025)
O1 (OPENAI ET AL., 2024B)
O3, O4-MINI (OPENAI, 2025B) |
| *Qwen*
QWEN3-VL 8B, 30B, 235B-A22B (QWEN TEAM, 2025) |

*Table 9.* Chess entity tracking with stronger metrics. **Exact**: exact-match FEN accuracy (%). **Changed**: accuracy on squares that changed from the initial position (%). **Legal**: fraction of predictions that are legal board positions (%). All values report mean ± 95% CI.

| | TEXT-ONLY | | | IMAGE-ONLY | | |
|---|---|---|---|---|---|---|
| MODEL | EXACT | CHANGED | LEGAL | EXACT | CHANGED | LEGAL |
| **BASELINE** | | | | | | |
| GAME START | 0.0 ± 0.0 | 0.0 ± 0.0 | 100.0 ± 0.0 | 0.0 ± 0.0 | 0.0 ± 0.0 | 100.0 ± 0.0 |
| **ZERO-SHOT** | | | | | | |
| GPT-4O | 2.0 ± 1.3 | 81.7 ± 0.8 | 58.6 ± 4.3 | 0.0 ± 0.4 | 14.1 ± 0.8 | 95.4 ± 1.9 |
| GPT-4O-MINI | 0.0 ± 0.4 | 25.2 ± 0.9 | 24.2 ± 3.7 | 0.0 ± 0.4 | 33.9 ± 1.0 | 21.8 ± 3.6 |
| GPT-4.1 | 1.2 ± 1.0 | 78.4 ± 0.9 | 66.4 ± 4.1 | 0.0 ± 0.4 | 25.8 ± 1.0 | 74.6 ± 3.8 |
| GPT-4.1-MINI | 0.4 ± 0.7 | 63.5 ± 1.1 | 35.8 ± 4.2 | 0.0 ± 0.4 | 15.4 ± 0.8 | 58.0 ± 4.3 |
| GPT-4.1-NANO | 0.0 ± 0.4 | 46.1 ± 1.1 | 51.0 ± 4.4 | 0.0 ± 0.4 | 3.4 ± 0.4 | 92.2 ± 2.4 |
| GEMINI-2.5-FLASH | 2.4 ± 1.4 | 83.6 ± 0.8 | 38.8 ± 4.3 | 0.0 ± 0.4 | 7.2 ± 0.6 | 59.6 ± 4.3 |
| LLAMA-4 MAVERICK | 0.0 ± 0.4 | 75.6 ± 0.9 | 40.4 ± 4.3 | 0.0 ± 0.4 | 48.2 ± 1.1 | 1.0 ± 0.9 |
| MINIMAX-VL-01 | 0.2 ± 0.5 | 67.9 ± 1.0 | 77.0 ± 3.7 | 0.0 ± 0.4 | 2.6 ± 0.4 | 94.0 ± 2.1 |
| CLAUDE 3.7 SONNET | 18.8 ± 3.4 | 90.0 ± 0.7 | 79.6 ± 3.5 | 0.0 ± 0.4 | 30.7 ± 1.0 | 82.2 ± 3.3 |
| **FEW-SHOT (N=5)** | | | | | | |
| GPT-4O | 3.2 ± 1.6 | 89.7 ± 0.7 | 83.6 ± 3.2 | 0.2 ± 0.5 | 33.5 ± 1.0 | 99.6 ± 0.7 |
| GPT-4O-MINI | 0.2 ± 0.5 | 47.1 ± 1.1 | 58.6 ± 4.3 | 0.0 ± 0.4 | 5.6 ± 0.5 | 63.0 ± 4.2 |
| GPT-4.1 | 2.4 ± 1.4 | 81.7 ± 0.8 | 79.6 ± 3.5 | 0.0 ± 0.4 | 29.6 ± 1.0 | 74.4 ± 3.8 |
| GPT-4.1-MINI | 0.6 ± 0.8 | 66.1 ± 1.0 | 70.4 ± 4.0 | 0.2 ± 0.5 | 34.6 ± 1.0 | 74.8 ± 3.8 |
| GPT-4.1-NANO | 0.0 ± 0.4 | 48.3 ± 1.1 | 50.4 ± 4.4 | 0.0 ± 0.4 | 0.7 ± 0.2 | 98.6 ± 1.1 |
| GEMINI-2.5-FLASH | 26.4 ± 3.9 | 90.2 ± 0.7 | 82.0 ± 3.4 | 0.2 ± 0.5 | 27.0 ± 1.0 | 89.0 ± 2.7 |
| LLAMA-4 MAVERICK | 0.4 ± 0.7 | 80.8 ± 0.9 | 66.8 ± 4.1 | 0.0 ± 0.4 | 36.0 ± 1.1 | 10.2 ± 2.7 |
| MINIMAX-VL-01 | 0.0 ± 0.4 | 68.7 ± 1.0 | 71.2 ± 4.0 | 0.0 ± 1.8 | 16.7 ± 1.8 | 54.0 ± 9.6 |
| CLAUDE 3.7 SONNET | 47.6 ± 4.4 | 98.5 ± 0.3 | 96.4 ± 1.7 | 0.2 ± 0.5 | 36.1 ± 1.1 | 75.6 ± 3.8 |
| **CHAIN-OF-THOUGHT** | | | | | | |
| GPT-4O | 7.8 ± 2.4 | 87.0 ± 0.7 | 85.0 ± 3.1 | 2.0 ± 1.3 | 61.4 ± 1.1 | 82.4 ± 3.3 |
| GPT-4O-MINI | 0.0 ± 0.4 | 11.3 ± 0.7 | 5.0 ± 1.9 | 0.0 ± 0.4 | 18.2 ± 0.8 | 11.6 ± 2.8 |
| GPT-4.1 | 42.8 ± 4.3 | 71.0 ± 1.0 | 69.0 ± 4.0 | 1.0 ± 0.9 | 24.3 ± 0.9 | 26.4 ± 3.9 |
| GPT-4.1-MINI | 5.0 ± 1.9 | 76.8 ± 0.9 | 54.2 ± 4.4 | 0.8 ± 0.9 | 56.1 ± 1.1 | 30.2 ± 4.0 |
| GPT-4.1-NANO | 0.0 ± 0.4 | 30.6 ± 1.0 | 32.2 ± 4.1 | 0.0 ± 0.4 | 3.8 ± 0.4 | 10.6 ± 2.7 |
| GEMINI-2.5-FLASH | 18.4 ± 3.4 | 69.9 ± 1.0 | 46.8 ± 4.4 | 5.2 ± 2.0 | 65.8 ± 1.0 | 34.2 ± 4.1 |
| LLAMA-4 MAVERICK | 10.6 ± 2.7 | 82.1 ± 0.8 | 64.2 ± 4.2 | 0.0 ± 0.4 | 33.6 ± 1.0 | 28.4 ± 3.9 |
| MINIMAX-VL-01 | 0.0 ± 0.4 | 52.7 ± 1.1 | 49.8 ± 4.4 | 0.0 ± 0.4 | 8.5 ± 0.6 | 26.2 ± 3.8 |
| CLAUDE 3.7 SONNET | 81.2 ± 3.4 | 98.9 ± 0.2 | 99.2 ± 0.9 | 30.0 ± 4.0 | 92.0 ± 0.6 | 97.0 ± 1.5 |
| **REASONING** | | | | | | |
| O1 | 61.0 ± 9.4 | 96.2 ± 0.9 | 90.0 ± 6.0 | 15.0 ± 7.0 | 62.0 ± 2.3 | 77.0 ± 8.2 |
| O3 | 99.0 ± 2.6 | 100.0 ± 0.1 | 99.0 ± 2.6 | 12.0 ± 6.4 | 36.6 ± 2.3 | 50.0 ± 9.6 |
| O4-MINI | 32.0 ± 9.0 | 82.4 ± 1.8 | 34.0 ± 9.1 | 6.0 ± 4.8 | 64.5 ± 2.3 | 37.0 ± 9.3 |
| GEMINI-2.5-PRO | 39.0 ± 9.4 | 75.5 ± 2.1 | 67.0 ± 9.1 | 49.0 ± 9.6 | 74.9 ± 2.1 | 69.0 ± 8.9 |
| GEMINI-2.5-FLASH | 17.0 ± 7.3 | 66.6 ± 2.3 | 52.0 ± 9.6 | 9.0 ± 5.7 | 55.6 ± 2.4 | 33.0 ± 9.1 |
| CLAUDE 3.7 SONNET | 88.0 ± 6.4 | 99.5 ± 0.4 | 99.0 ± 2.6 | 52.0 ± 9.6 | 93.5 ± 1.2 | 95.0 ± 4.5 |
| CLAUDE 4.5 OPUS | 97.0 ± 3.7 | 100.0 ± 0.1 | 100.0 ± 1.8 | 89.0 ± 6.2 | 99.0 ± 0.5 | 100.0 ± 1.8 |
| GPT-5.2 | 95.0 ± 4.5 | 98.0 ± 0.7 | 98.0 ± 3.2 | 88.0 ± 6.4 | 98.5 ± 0.6 | 100.0 ± 1.8 |

