# OpenReview forum: "MET-Bench: Multimodal Entity Tracking for Evaluating the Limitations of Vision-Language and Reasoning Models"
_ICML.cc/2026/Conference — ICML 2026 regular_

### Official Review · Reviewer_XQAs · 2026-03-09

**Soundness:** 2
**Presentation:** 3
**Significance:** 2
**Originality:** 3
**Overall Recommendation:** 4
**Confidence:** 3

**Summary:**

This paper presents MET-Bench, a benchmark for multimodal entity tracking, using Chess and Shell Game to evaluate whether models can update and maintain state when actions are provided in either text or image form. Experiments across a range of language and vision-language models show a consistent performance gap between text-based and image-based tracking. The paper also includes diagnostic analyses, including action recognition, cascaded inference, long-horizon evaluation, and GRPO-based training, to study the sources of error in multimodal state tracking.

**Compliance With Llm Reviewing Policy:**

Affirmed.

**Final Justification:**

The authors clarified my concern in the rebuttal.

**Key Questions For Authors:**

1. In the RL experiments, did you compare GRPO against simple supervised fine-tuning on solved trajectories or distilled chain-of-thought traces? If not, it is hard to know whether the RL component is doing something special versus just additional task-specific adaptation.

2. Can you report stronger Chess metrics, such as exact-match FEN accuracy, changed-square accuracy, and optionally legal-board validity, at least for the main models in Table 1 and the sequence-length analysis in Figure 3? The current per-square metric appears relatively forgiving.

3. Since this is a benchmark paper, readers need exact split sizes, generation details, and ideally a clear release plan. Without this, the work reads more like an internal evaluation suite than a community benchmark.

**Limitations:**

The paper includes a brief impact statement, but the discussion of limitations is quite limited. It would help to more explicitly acknowledge the narrow benchmark scope and the limited mixed-modality evaluation in the main paper.

**Strengths And Weaknesses:**

Strengths

1. The paper studies an important and relatively underexplored problem: multimodal entity tracking under sequential state updates.

2. The benchmark design is clean and easy to follow, and the paper includes useful diagnostics beyond headline accuracy, such as long-horizon evaluation, action recognition, cascaded inference, and RL-based adaptation.

Weaknesses

1. The paper’s framing emphasizes multimodal entity tracking, but the main empirical evidence is more directly about a text-versus-image performance gap than about mixed-modality integration itself. The core experiments in Section 4.1 compare trajectories where actions are provided entirely in text or entirely in images, and the main tables are organized around this contrast. By comparison, evidence on interleaved text-image tracking appears only later in the appendix (Figure 6), so support for the broader cross-modal integration claim is relatively limited in the main paper.

2. The Chess metric is somewhat forgiving for the conclusions drawn from it. The paper uses per-square board accuracy and notes that simply predicting the initial board is already a strong baseline, since much of the board remains unchanged after ten actions. In Table 1, this baseline is already 74.9 ± 0.5. As a result, partially incorrect trajectories can still receive relatively high scores, which weakens the interpretation of reported Chess performance differences.

3. The paper’s reasoning-versus-perception conclusion is stronger than the evidence fully supports, especially for Chess. The paper argues that the main bottleneck lies in sequential reasoning rather than visual recognition, but Table 4 shows substantial Chess action-recognition error for some models, including 46.4 ± 1.0 for GPT-4o-mini and 67.5 ± 2.9 for Gemma3 27B. The evidence is much cleaner for Shell Game, where action recognition is essentially perfect, so this conclusion feels better supported there than in Chess.

4. The RL section is interesting but too limited to support strong general claims. GRPO is only applied on a 5-action version of the benchmark, and the paper itself reports limited or no cross-modal transfer. As presented, this reads more like a promising pilot experiment than a fully convincing demonstration that RL materially solves multimodal entity tracking.

---

> ### Author Rebuttal · Authors · 2026-03-30
>
> ### We thank the reviewer for the specific suggestions. We respond with new results:
>
> ---
>
> ## W1: Framing Emphasizes Multimodality but Main Evidence is Text-vs-Image; Interleaved Only in Appendix
>
> We will clarify these points further in the paper, and make clear the nature of the multimodality. We note that the image-action condition is inherently multimodal: the model receives state as text, observes actions as images, and predicts state as text, integrating across modalities within one inference. The primary goal of MET-Bench is to compare entity tracking between the multimodal case (image-only actions and text states) and the text-only case. If the reviewer believes the mixed-modality ablation deserves prominence, we will promote these results to the main paper.
>
> ## W2 + Q2: Stronger Chess Metrics:
>
> We have implemented the suggested metrics. In the table below, the strongest models attain high accuracy across all metrics; per-square accuracy was not masking poor performance. But GPT-4o drops from 94.9% per-square to 13.4% exact-match in text-only, illustrating that per-square accuracy can be forgiving for weaker models. The text-to-image performance gap is preserved under all four metrics, so our conclusions remain unchanged.
>
> | Model | Mode | Per-Sq | Exact | Changed | Legal |
> |-------|------|--------|-------|---------|-------|
> | GPT-4o | text-only | 94.9 ± 0.2 | 13.4 ± 3.0 | 89.3 ± 0.7 | 88.2 ± 2.8 |
> | GPT-4o | image-only | 67.4 ± 0.5 | 0.2 ± 0.5 | 50.0 ± 1.1 | 73.0 ± 3.9 |
> | GPT-5.2 | text-only | 98.2 ± 0.2 | 95.8 ± 2.4 | 98.1 ± 0.4 | 97.5 ± 1.9 |
> | GPT-5.2 | image-only | 98.8 ± 0.1 | 89.0 ± 2.7 | 98.2 ± 0.3 | 99.4 ± 0.8 |
> | Gemini 3 Pro | text-only | 96.7 ± 0.2 | 88.6 ± 2.8 | 95.5 ± 0.5 | 93.4 ± 2.2 |
> | Gemini 3 Pro | image-only | 95.6 ± 0.2 | 77.0 ± 3.7 | 92.5 ± 0.6 | 89.6 ± 2.7 |
> | Claude Opus 4.5 | text-only | 100.0 ± 0.0 | 99.0 ± 0.9 | 100.0 ± 0.0 | 100.0 ± 0.4 |
> | Claude Opus 4.5 | image-only | 99.5 ± 0.1 | 88.6 ± 4.6 | 99.0 ± 0.4 | 100.0 ± 1.0 |
>
> ## W3: Perception vs. Reasoning Overstated for Chess
>
> The reviewer correctly identifies that perception remains a meaningful bottleneck for weaker models on Chess (but not Shell Game). GPT 4o mini and Gemma3 27B cannot fully decouple perception from reasoning. However, for models with strong Chess perception (GPT 4o at 93%, GPT 5.4 Mini at 99.5%), our cascaded analysis shows that 82 to 100% of the text to image gap is attributable to visual reasoning rather than perception. We will restructure the discussion to present this as a spectrum: perception is the first barrier, and once cleared, visual reasoning remains a bottleneck.
>
> | Pipeline | GPT-4o (93% classif.) | GPT-5.4 Mini (99.5% classif.) |
> |----------|----------------------|------------------------------|
> | Text-Only | 94.9 ± 0.2 | 92.4 ± 0.3 |
> | Cascaded | 93.2 ± 0.2 | 92.5 ± 0.3 |
> | Image-Only | 67.5 ± 0.5 | 89.1 ± 0.3 |
>
> ---
>
> ## W4: RL Section Too Limited
>
> To address the reviewer's concern, we trained Gemma3 4B on text-only action sequences of length 10 in Shell. Accuracy on the 10-action test set improves from 36% to 47% (+11) for text and 33% to 42% (+9) for image. We will fully update the RL section with improved 10-action experiments from the 5-action version.
>
> The RL experiments are meant as a diagnostic. We show how performance in-modality can be improved through RL reasoning training. The absence of cross modal transfer is, to our knowledge, the first empirical evidence that VLM entity tracking operates through modality specific pathways rather than shared representations. This finding is itself a contribution, as it directly informs architectural choices for future VLM design.
>
> ## Q1: GRPO vs. SFT
>
> We find SFT yields little to no improvement with either text or image observations in our setup. The model can learn specific sequences, but fails to generalize to held out evaluations. We will include these findings in our revised paper. Solving the state tracking tasks is easier with a CoT scratchpad and training CoT requires RL. Distillation from stronger models is an interesting direction but orthogonal to our goal of demonstrating that RL can improve entity tracking without privileged access to stronger models' reasoning traces.
>
> ## Q3: Benchmark Release
>
> We will release the full benchmark on HuggingFace Hub and GitHub, including all generated game trajectories, image renderings, evaluation scripts, and data generation code. Evaluation code will reproduce all tables in the paper. The full generated datasets (train/val/test) are: Chess (99K, 2K, 3K), Shell: (50K, 1K, 5K), and Minecraft (5K/500/1.5K). We utilize a N=500 subset of Chess and Shell for evaluation (while reporting 95% CIs).
>
> ---
>
> ## Limitations:
> We will include an expanded limitation section discussing the referenced points.
>
> ### We hope the concerns identified in the review have been substantively addressed and welcome further discussion.

---

> > ### Author Rebuttal · Reviewer_XQAs · 2026-04-04
> >
> > Thanks the authors for the reply. The authors have clarified my concern in the rebuttal.

---

### Official Review · Reviewer_bdW5 · 2026-03-11

**Soundness:** 3
**Presentation:** 3
**Significance:** 2
**Originality:** 3
**Overall Recommendation:** 4
**Confidence:** 3

**Summary:**

This paper introduces MET-Bench, a benchmark consisting of multimodal chess and shell game tasks, to evaluate the ability of vision-language models (VLMs) to track entity states across modalities. The study reveals a substantial performance gap between text-based and image-based tracking, which is mainly caused by insufficient continuous visual reasoning rather than deficiencies in basic perception. Although reinforcement learning (GRPO) can effectively improve the reasoning performance of open-source models and bring them to a level comparable to closed-source models, the lack of cross-modal transferability suggests that future research should explore explicit memory structures and novel multimodal representation learning methods to bridge the gap between textual and visual entity tracking.

**Compliance With Llm Reviewing Policy:**

Affirmed.

**Final Justification:**

The authors have addressed my concerns, and I maintain my rating.

**Key Questions For Authors:**

see weakness

**Limitations:**

yes

**Strengths And Weaknesses:**

**Strengths:**

**Deep mechanistic insights that challenge common assumptions:**
Rather than stopping at the superficial conclusion that the model performs poorly on image-based tasks, the paper goes further and shows through image-action classification experiments that the model is actually able to perceive and classify individual visual actions accurately.

This clearly indicates that the main bottleneck of current VLMs does not lie in visual perception itself, but in implicit state reconstruction and sequential reasoning over continuous visual observations.

**A clear roadmap for future research:**
Through chain-of-thought (CoT) and cascaded reasoning experiments, the paper demonstrates the strong potential of explicit reasoning.

At the same time, it honestly reveals that current RL training lacks cross-modal transferability, suggesting that the architecture is still learning modality-specific representations rather than truly shared reasoning mechanisms. This directly motivates future work on explicit memory structures, cross-modal alignment objectives, and mechanistic interpretability methods.

---

**Weaknesses:**

**Evaluation limited to highly structured and idealized environments:**
The MET-Bench benchmark used in this paper, including multimodal chess and shell-game tasks, is built on highly discrete, rule-based, and controlled environments.

**Lack of discussion of related domains and literature:**
The paper does not sufficiently discuss several relevant research areas and prior works, such as natural language tracking developed from visual tasks (e.g., TNL2K[1] and MGIT[2]), as well as instance segmentation methods such as SAM[3] and SAM2[4].

**Insufficient experimental coverage:**
The experiments should include a broader range of LLMs and VLMs, such as the Qwen series, Qwen-VL series, and LLaVA series.

[1] Wang, Xiao, et al. "Towards more flexible and accurate object tracking with natural language: Algorithms and benchmark." Proceedings of the IEEE/CVF conference on computer vision and pattern recognition. 2021.
[2] Hu, Shiyu, et al. "A multi-modal global instance tracking benchmark (mgit): Better locating target in complex spatio-temporal and causal relationship." Advances in neural information processing systems 36 (2023): 25007-25030.
[3] Kirillov, Alexander, et al. "Segment anything." Proceedings of the IEEE/CVF international conference on computer vision. 2023.
[4] Ravi, Nikhila, et al. "Sam 2: Segment anything in images and videos." arXiv preprint arXiv:2408.00714 (2024).

---

> ### Author Rebuttal · Authors · 2026-03-30
>
> ### We thank the reviewer for the detailed feedback. We respond with new experiments:
>
> ---
>
> ## W1: Evaluation Limited to Highly Structured and Idealized Environments
>
> To address the reviewer's feedback and bridge the gap between realistic and structured domains, we add a third domain: Minecraft world-state prediction. Unlike Chess and Shell Game where text and image represent the same actions, Minecraft tests world modeling when the state is visual versus symbolic. These tasks involve partial observation, dynamic environments, and visually complex scenes. Given a current state and a text action (e.g., "move forward, then attack"), the model selects the correct next state from four candidates. In the text condition, states are scene graphs; in the image condition, first-person rendered frames. Evaluations are on 1,500 multiple choice questions generated from 90 diverse trajectories.
>
> MET-Bench's controlled domains are an important feature. Exact ground-truth state enables task decomposition and modality comparison (cascaded inference, action-perception probes) that would be infeasible in naturalistic settings. Structured tasks allow systematic difficulty scaling via sequence length.
>
> | Model | Text Acc | Image Acc |
> |-------|----------|-----------|
> | Random | 25.0 | 25.0 |
> | Gemma-3 4B | 45.0 ± 4.3 | 27.0 ± 3.9 |
> | Gemma-3 27B | 53.2 ± 4.4 | 30.4 ± 4.0 |
> | Qwen3-VL 8B | 54.0 ± 4.3 | 26.4 ± 3.9 |
> | Qwen3-VL 30B | 72.0 ± 3.9 | 31.4 ± 4.1 |
> | Claude Sonnet 4.6 | 88.4 ± 2.8 | 41.8 ± 4.3 |
>
> The text-image gap persists whether vision is required to parse actions (Chess/Shell) or to represent states (Minecraft), suggesting VLMs struggle with visual reasoning broadly rather than visual action parsing specifically.
>
> ---
>
> ## W2: Lack of Discussion of Related Domains and Literature (TNL2K, MGIT, SAM, SAM2)
>
> We will add discussion of these works and clarify the distinction. TNL2K and MGIT evaluate visual object tracking with natural language guidance, where the task is spatial localization: predicting bounding boxes that follow a specified object across video frames. SAM and SAM2 address segmentation, identifying pixel-level object boundaries given prompts.
>
> MET-Bench evaluates a different capability: symbolic state inference from action sequences. The task is to determine the world state after a series of actions. Our Chess and Shell Game domains have trivial localization (fixed board positions, three labeled shells) yet models still fail, precisely because the bottleneck is reasoning about state updates rather than spatial perception.
>
> ---
>
> ## W3: Insufficient Experimental Coverage (Qwen, LLaVA)
>
> We have evaluated additional model families as the reviewer suggested. The results confirm the text to image gap generalizes across more models:
>
> #### Chess (Square Accuracy, 95% CI)
>
> | Model | Image-Only | Text-Only | Text−Image Gap |
> |-------|------------|-----------|----------------|
> | Qwen2.5-VL-32B | 54.75% ± 1.22% | 67.25% ± 1.15% | +12.5pp |
> | Mistral Small 3.2 24B | 29.41% ± 1.12% | 63.25% ± 1.18% | +33.8pp |
>
> #### Shell Game (Accuracy, 95% CI)
>
> | Model | Image-Only | Text-Only | Text−Image Gap |
> |-------|------------|-----------|----------------|
> | Qwen2.5-VL-32B | 29.00% ± 8.76% | 93.00% ± 5.16% | +64.0pp |
> | Mistral Small 3.2 24B | 44.00% ± 9.55% | 90.00% ± 5.96% | +46.0pp |

---

> > ### Author Rebuttal · Reviewer_bdW5 · 2026-04-03
> >
> > Thank you to the authors for the additional experiments and clarifications. I encourage the authors to incorporate these supplementary results and discussions into the final version of the paper. I have no further questions, and I will maintain my positive rating.

---

### Official Review · Reviewer_2j6b · 2026-03-13

**Soundness:** 2
**Presentation:** 3
**Significance:** 3
**Originality:** 3
**Overall Recommendation:** 4
**Confidence:** 4

**Summary:**

This paper introduces MET-Bench, a benchmark for multimodal entity tracking, with two structured domains: Chess and Shell Game. The task is to predict the final entity state from an initial text state and a sequence of actions, where actions are provided either as text or as images. The paper shows that current VLMs perform much better when state updates are presented in text than in images, even when image-level action perception is relatively strong. It further argues, via image-action classification and cascaded inference, that the main bottleneck is not simple perception but reasoning over visual state updates. Finally, the paper applies GRPO-based reinforcement learning to open-source VLMs and shows that RL can improve in-modality entity tracking, although cross-modal transfer remains limited. Overall, the paper presents a new benchmark, a careful empirical study of multimodal entity tracking, and an initial RL-based improvement strategy.

**Compliance With Llm Reviewing Policy:**

Affirmed.

**Key Questions For Authors:**

1.Generalization: Could the authors discuss or provide preliminary insights on how the findings from these discrete, synthetic domains might map onto continuous, noisy video tracking tasks?

2.Perception Bottleneck: Given that perception is not flawless in the Chess domain, could you tone down the absolute decoupling of perception and reasoning, or provide a deeper analysis of failure cases where imperfect perception explicitly derails reasoning?

3.RL Setup: Why was the RL training limited to the 5-action version? Furthermore, do the authors have any intuition or future directions on how to achieve the missing cross-modal transfer using RL in this setting?

4.Positioning: Please explicitly clarify what distinguishes the core challenges of MET-Bench from existing multimodal procedural state-understanding tasks in the related work section.

**Limitations:**

The authors have included a dedicated discussion on limitations, which is appreciated. Nonetheless, I encourage the authors to be more explicit about the limited ecological validity of MET-Bench. The current domains are highly structured, discrete, and visually clean, which inherently restricts their ability to represent the complexities of real-world entity tracking (e.g., continuous natural videos, cluttered scenes).

**Strengths And Weaknesses:**

**Strengths**

Elegant Methodological Design: The benchmark design is exceptionally clean. Repurposing structured games to isolate multimodal state tracking—specifically by contrasting end-to-end image tracking with cascaded inference—is a clever and effective way to decouple perception from reasoning.

Rigorous Empirical Analysis: The experimental suite is strong and convincing. The authors evaluate a wide spectrum of frontier models across diverse settings (zero/few-shot, CoT, varying sequence lengths) and successfully use action-classification accuracy probes to substantiate their central claims.

Valuable Insights for the Community: The explicit demonstration that visual reasoning, rather than merely visual perception, is the limiting factor in long-horizon multimodal world modeling is a significant finding that will benefit future research in robotics, agentic AI, and video QA.

Clarity and Presentation: The paper is highly readable and well-structured. The narrative flows logically from benchmark introduction to empirical diagnosis, and finally to the RL-based intervention.

**Weaknesses**

Limited External Validity (Synthetic Domains): Both Chess and the Shell Game are highly structured environments with discrete state updates and clean visual renderings. While excellent for controlled diagnosis, it is unclear how strongly these conclusions will transfer to more realistic, open-world multimodal tracking scenarios (e.g., cluttered scenes, continuous natural videos).

Overstated "Perception vs. Reasoning" Dichotomy: The claim that the performance gap is primarily due to reasoning is plausible, but the boundary might not be fully closed. Image understanding in the full task likely requires more than just recognizing an isolated action symbol, particularly in the Chess domain where low-level perception is not uniformly perfect across all baseline models.

Constrained Scope of the RL Intervention: The RL contribution, while promising, feels somewhat preliminary. Training is restricted to a simplified 5-action version of the benchmark and evaluated only on the Gemma-3 (4B/27B) models. More importantly, the lack of cross-modal transfer limits the practical impact of this intervention—showing that RL helps within a modality but does not yet bridge the multimodal reasoning gap.

Insufficient Contextualization in Related Work: The paper would benefit from a sharper discussion differentiating MET-Bench from adjacent evaluation settings, such as existing video QA benchmarks, multimodal procedural state tracking, or other world-model evaluations.

---

> ### Author Rebuttal · Authors · 2026-03-30
>
> ### We thank the reviewer for the thoughtful feedback. We respond with new experiments:
>
> ---
>
> ## W1: Limited External Validity (Synthetic Domains) / Q1: Generalization to Continuous, Noisy Video Tracking
>
> To address the reviewer's feedback and bridge the gap between realistic and structured domains, we add a third domain: Minecraft world-state prediction. Unlike Chess and Shell Game where text and image represent the same actions, Minecraft tests world modeling when the state is visual versus symbolic. These tasks involve partial observation, dynamic environments, and visually complex scenes. Given a current state and a text action (e.g., "move forward, then attack"), the model selects the correct next state from four candidates. In the text condition, states are scene graphs; in the image condition, first-person rendered frames. Evaluations are on 1,500 multiple choice questions generated from 90 diverse trajectories.
>
> MET-Bench's controlled domains are an important feature. Exact ground-truth state enables task decomposition and modality comparison (cascaded inference, action-perception probes) that would be infeasible in naturalistic settings. Structured tasks allow systematic difficulty scaling via sequence length.
>
> | Model | Text Acc | Image Acc |
> |-------|----------|-----------|
> | Random | 25.0 | 25.0 |
> | Gemma-3 4B | 45.0 ± 4.3 | 27.0 ± 3.9 |
> | Gemma-3 27B | 53.2 ± 4.4 | 30.4 ± 4.0 |
> | Qwen3-VL 8B | 54.0 ± 4.3 | 26.4 ± 3.9 |
> | Qwen3-VL 30B | 72.0 ± 3.9 | 31.4 ± 4.1 |
> | Claude Sonnet 4.6 | 88.4 ± 2.8 | 41.8 ± 4.3 |
>
> The text-image gap persists whether vision is required to parse actions (Chess/Shell) or to represent states (Minecraft), suggesting VLMs struggle with visual reasoning broadly rather than visual action parsing specifically.
>
> ---
>
> ## W2: Overstated "Perception vs. Reasoning" Dichotomy / Q2: Deeper Failure Case Analysis
>
> Thank you, we will qualify that weaker models face larger perception limits in Chess, but the core finding holds for competitive VLMs.
>
> We run the cascaded decomposition on two models with high chess move classification accuracy: GPT-4o (93.0% per-action) and (New) GPT-5.4 Mini (99.5% per-action). The cascaded pipeline feeds VLM-extracted text actions to the same model for end-to-end board prediction. If perception were the bottleneck, cascaded accuracy would remain low; if visual reasoning were the bottleneck, cascaded would recover to text-only levels. Across both models, 82–100% of the text-image gap is visual reasoning. We will restructure the discussion to present this as a spectrum: perception is the first barrier, and once cleared, visual reasoning remains a bottleneck.
>
> | Pipeline | GPT-4o (93% classif.) | GPT-5.4 Mini (99.5% classif.) |
> |----------|----------------------|------------------------------|
> | Text-Only | 94.9 ± 0.2 | 92.4 ± 0.3 |
> | Cascaded | 93.2 ± 0.2 | 92.5 ± 0.3 |
> | Image-Only | 67.5 ± 0.5 | 89.1 ± 0.3 |
>
> ---
>
> ## W3: Constrained Scope of the RL Intervention / Q3: Why Limited to 5-Action? Cross-Modal Transfer Directions?
>
> To address the reviewer's concern, we trained Gemma3 4B on text-only action sequences of length 10 in Shell. Accuracy on the 10-action test set improves from 36% to 47% (+11) for text and 33% to 42% (+9) for image. We will fully update the RL section with improved 10-action experiments from the 5-action version.
>
> The RL experiments serve as a diagnostic. We show how performance in-modality can be improved through RL reasoning training. The absence of cross modal transfer is, to our knowledge, the first empirical evidence that VLM entity tracking operates through modality specific pathways rather than shared representations. We believe the empirical finding that RL training produces modality specific rather than shared reasoning pathways is itself a contribution, as it directly informs architectural choices for future VLM design.
>
> GRPO training improves open-source models to accuracy competitive with closed models.
>
> ---
>
> ## W4: Insufficient Contextualization in Related Work / Q4: Differentiation from Existing Benchmarks
>
> We will expand the related work to address the categories the reviewer raises.
>
> MET-Bench's core differentiator is its parallel modality design: identical tasks with text and image conditions, exact ground truth states enabling cascaded decomposition, and action perception probes that isolate perception from reasoning. No existing benchmark combines these three properties. Video QA benchmarks lack symbolic ground truth state. Procedural state tracking benchmarks lack parallel modality conditions. World model evaluations typically focus on generation quality rather than symbolic state accuracy.
>
> Beyond diagnosis, MET-Bench's structured domains allow automatic difficulty scaling through sequence length or state complexity as models improve. We will add this positioning to the revision.

---

### Official Review · Reviewer_iZhg · 2026-03-13

**Soundness:** 2
**Presentation:** 2
**Significance:** 1
**Originality:** 2
**Overall Recommendation:** 4
**Confidence:** 4

**Summary:**

This paper presents MET Bench, a benchmark for multimodal entity tracking in Chess and Shell Game. The setup gives the initial state in text, a sequence of actions in either text or images, and asks the model to predict the final state in text. The main empirical result is a large gap between text action tracking and image action tracking across many models. The paper further argues that this gap is more about sequential reasoning than simple perception, based on single action classification and a cascaded image to text pipeline. It also shows that GRPO training can improve open models within one modality, but transfer across modalities is limited.

**Compliance With Llm Reviewing Policy:**

Affirmed.

**Final Justification:**

Score 2 -> 4 after rebuttal

**Key Questions For Authors:**

Please see weaknesses section.

**Limitations:**

Please see weaknesses section.

**Strengths And Weaknesses:**

**Strengths**

- The paper asks a clear question and the task setup is easy to understand and evaluate.

- The benchmark is controlled, so it is easy to isolate one capability, namely state tracking under sequential updates.

- The experiments cover many models and several prompting settings, and the long sequence results are useful for showing error accumulation.

- The cascaded analysis is a nice diagnostic and helps separate direct image based tracking from text based tracking after transcription.

**Weaknesses**

- The main weakness is that the benchmark is too narrow and too synthetic for the broad claims the paper wants to make. Both tasks are small toy domains with artificial visuals and symbolic outputs. Because of that, the benchmark looks more like a diagnostic probe than a convincing benchmark for real multimodal entity tracking or world modeling. The paper does not give enough grounding to argue that failure on these two tasks reflects practical multimodal reasoning ability in realistic settings.

- The claim that the gap mainly comes from reasoning rather than perception feels stronger than the evidence. The single action classification results only show that models can often read isolated actions, not that perception is no longer an important bottleneck during sequential state updates. The cascaded experiment is helpful, but it is still limited and does not fully justify such a strong conclusion across both domains and across models.

- The reinforcement learning section feels smaller than the way it is framed. The training setup is quite limited, and the evidence mostly supports in modality improvement on short tasks. That is interesting, but it is not enough to strongly support the broader claim that the method reaches performance comparable to advanced closed models in a meaningful benchmark sense.

---

> ### Author Rebuttal · Authors · 2026-03-30
>
> ### We thank the reviewer for the feedback. We respond with new results:
> ---
>
> ## W1: Benchmark Too Narrow and Synthetic for Broad Claims
>
> The reviewer characterizes MET-Bench as "more like a diagnostic probe than a convincing benchmark." MET-Bench follows the tradition of diagnostic benchmarks that have driven significant progress: CLEVR [1] for compositional reasoning, ViLP [2] for visual language priors, and GSM-Symbolic [3] for mathematical reasoning. Each uses synthetic, controlled settings precisely because naturalistic benchmarks cannot isolate the specific failure modes they target. MET-Bench applies this methodology to multimodal entity tracking for the first time.
>
> To address the reviewer's feedback and bridge the gap between realistic and structured domains, we add a third domain: Minecraft world state prediction. Unlike Chess and Shell Game, Minecraft involves partial observation, dynamic 3D environments, and visually complex rendered scenes. Given a current state and a text action, the model selects the correct next state from four candidates. Evaluations are on 1,500 multiple choice questions generated from 90 diverse trajectories.
>
> | Model | Text Acc | Image Acc |
> |-------|----------|-----------|
> | Random | 25.0 | 25.0 |
> | Gemma 3 4B | 45.0 ± 4.3 | 27.0 ± 3.9 |
> | Gemma 3 27B | 53.2 ± 4.4 | 30.4 ± 4.0 |
> | Qwen3 VL 8B | 54.0 ± 4.3 | 26.4 ± 3.9 |
> | Qwen3 VL 30B | 72.0 ± 3.9 | 31.4 ± 4.1 |
> | Claude Sonnet 4.6 | 88.4 ± 2.8 | 41.8 ± 4.3 |
>
> The text-image gap persists whether vision is required to parse actions (Chess/Shell) or to represent states (Minecraft), suggesting VLMs struggle with visual reasoning broadly rather than visual action parsing specifically.
>
> [1] Johnson et al., "CLEVR: A Diagnostic Dataset for Compositional Language
> and Elementary Visual Reasoning," CVPR 2017.
> [2] Luo et al., "Probing Visual Language Priors in VLMs," ICML 2025.
> [3] Mirzadeh et al., "GSM-Symbolic: Understanding the Limitations of
> Mathematical Reasoning in Large Language Models," ICLR 2025.
>
> ---
>
> ## W2: Perception vs. Reasoning Claim Stronger Than the Evidence
>
> The reviewer states that single action classification "only shows that models can often read isolated actions, not that perception is no longer an important bottleneck during sequential state updates." The cascaded experiment directly tests perception vs. visual reasoning. If perception errors were compounding during sequential updates, the cascaded pipeline (which removes the need to perceive during tracking) would not recover the performance of the text-only task.
>
> In Shell Game, all evaluated models attain 100% image-action classification accuracy, so image-only failures are unambiguously reasoning failures. This level of attribution is only possible because MET-Bench provides exact ground-truth state in a controlled domain.
>
> We will qualify that weaker models face larger perception limits in Chess, but the core finding holds for competitive VLMs. We will restructure the discussion to present this as a spectrum: perception is the first barrier, and once cleared, visual reasoning remains a bottleneck.
>
> We run the cascaded decomposition on two models with high chess move classification accuracy: GPT-4o (93.0% per-action) and (New) GPT-5.4 Mini (99.5% per-action). The cascaded pipeline feeds VLM-extracted text actions to the same model for end-to-end board prediction. Across both models, 82–100% of the text-image gap is visual reasoning.
>
> | Pipeline | GPT-4o (93% classif.) | GPT-5.4 Mini (99.5% classif.) |
> |----------|----------------------|------------------------------|
> | Text-Only | 94.9 ± 0.2 | 92.4 ± 0.3 |
> | Cascaded | 93.2 ± 0.2 | 92.5 ± 0.3 |
> | Image-Only | 67.5 ± 0.5 | 89.1 ± 0.3 |
>
> ## W3: Reinforcement Learning Section Too Preliminary
>
> To address the reviewer's concern, we trained Gemma3 4B on text-only action sequences of length 10 in Shell. Accuracy on the 10-action test set improves from 36% to 47% (+11) for text and 33% to 42% (+9) for image. We will fully update the RL section with improved 10-action experiments from the 5-action version.
>
> The key finding is that RL produces modality-specific reasoning with limited cross-modal transfer, directly informing VLM architecture design. This is, to our knowledge, the first empirical evidence that VLM entity tracking operates through modality specific pathways rather than shared representations. We believe this is itself a contribution, independent of whether RL fully solves the task, as it directly informs architectural choices for future VLM design.
>
> ---
>
> ### Given these additions, we believe the concerns identified in the review have been substantively addressed and welcome further discussion.

---

> > ### Author Rebuttal · Reviewer_iZhg · 2026-04-03
> >
> > I appriciate the authors' rebuttal and the additional settings and experiments. My concerns have been resolved and will raise score to 4.

---

### Decision · Program_Chairs · 2026-04-30

**Decision:**

Accept (regular)

**Comment:**

The reviewers are in strong agreement (all weak accept) that this paper presents a clear and well-executed diagnostic study of multimodal entity tracking. The benchmark is cleanly designed, and the empirical analysis—particularly the use of controlled settings and diagnostic experiments—provides useful insights into the gap between text-based and image-based tracking in current VLMs.

The main concerns are consistent across reviewers: the benchmark is relatively narrow and synthetic, which limits external validity; the reasoning-versus-perception claim is somewhat overstated; and the RL component is preliminary with limited cross-modal impact. Additional issues include metric limitations and limited emphasis on mixed-modality settings.

The rebuttal effectively addresses most concerns by adding experiments and clarifying claims, and reviewers explicitly acknowledge that their issues have been resolved.

Overall, despite some limitations in scope and positioning, the paper offers a solid and useful contribution as a diagnostic benchmark and analysis tool, and meets the bar for acceptance.